# Lemongrass Essential Oil Components with Antimicrobial and Anticancer Activities

**DOI:** 10.3390/antiox11010020

**Published:** 2021-12-22

**Authors:** Mohammad Mukarram, Sadaf Choudhary, Mo Ahamad Khan, Palmiro Poltronieri, M. Masroor A. Khan, Jamin Ali, Daniel Kurjak, Mohd Shahid

**Affiliations:** 1Advance Plant Physiology Section, Department of Botany, Aligarh Muslim University, Aligarh 202002, India; sadafchoudhary92@gmail.com (S.C.); mmakhan.bt@amu.ac.in (M.M.A.K.); 2Department of Integrated Forest and Landscape Protection, Faculty of Forestry, Technical University in Zvolen, T. G. Masaryka 24, 96001 Zvolen, Slovakia; kurjak@tuzvo.sk; 3Department of Microbiology, Jawaharlal Nehru Medical College, Aligarh Muslim University, Aligarh 202002, India; ma.khan@myamu.ac.in; 4Institute of Sciences of Food Productions, ISPA-CNR, National Research Council of Italy, Via Monteroni km 7, 73100 Lecce, Italy; 5Centre for Applied Entomology and Parasitology, School of Life Sciences, Keele University, Keele, Newcastle ST5 5BG, UK; j.ali@keele.ac.uk; 6Department of Microbiology, Immunology & Infectious Diseases, College of Medicine and Medical Sciences, Arabian Gulf University, Road 2904 Building 293 Manama, 329, Bahrain; mohammeds@agu.edu.bh

**Keywords:** anticancer, antimicrobial, antioxidants, cancer signalling, citral, *Cymbopogon*, essential oil

## Abstract

The prominent cultivation of lemongrass (*Cymbopogon* spp.) relies on the pharmacological incentives of its essential oil. Lemongrass essential oil (LEO) carries a significant amount of numerous bioactive compounds, such as citral (mixture of geranial and neral), isoneral, isogeranial, geraniol, geranyl acetate, citronellal, citronellol, germacrene-D, and elemol, in addition to other bioactive compounds. These components confer various pharmacological actions to LEO, including antifungal, antibacterial, antiviral, anticancer, and antioxidant properties. These LEO attributes are commercially exploited in the pharmaceutical, cosmetics, and food preservations industries. Furthermore, the application of LEO in the treatment of cancer opens a new vista in the field of therapeutics. Although different LEO components have shown promising anticancer activities in vitro, their effects have not yet been assessed in the human system. Hence, further studies on the anticancer mechanisms conferred by LEO components are required. The present review intends to provide a timely discussion on the relevance of LEO in combating cancer and sustaining human healthcare, as well as in food industry applications.

## 1. Introduction

Cymbopogon spp. are fast-growing C_4_ perennial sedges from the grass family Poaceae and are primarily cultivated for their essential oils. The genus lemongrass comprises about 180 species, such as *Cymbopogon citratus*, *Cymbopogon flexuosus*, *Cymbopogon winterianus*, *Cymbopogon martinii*, *Cymbopogon nardus*, and *Cymbopogon refractus*. These aromatic grasses are of great commercial interest due to their wide applications in different areas such as the food, pharmaceutical, and cosmetic industries. The plant propagates through seed and slips and has thin and lanceolate leaves that appear to emerge directly from the soil without any stem [1]. Although lemongrass cultivation is cosmopolitan, India has a monopoly over its production and export [2,3]. Given the growing consumer demand and global market for essential oil, few recent reports have suggested various sustainable approaches for enhancing lemongrass production even further, either in field condition or in culture, through the application of elicitors, nanoparticles, and metabolite regulators [3,4].

Lemongrass is also called Cochin grass, since 90% of its global export is organised from Cochin port, India [5]. Lemongrass plant has extensive potential as food and fodder given its richness in vitamins A, C, E, folate, niacin, and riboflavin, protein, antioxidants, and mineral nutrients, such as N (0.74%), P (0.07%), K (2.12%), S (0.19%), Mg (0.15%), Ca (0.36%), Zn (35.51 ppm), Mn (155.82%), Fe (126.73%), and Cu (56.64 ppm) [6,7]. Figure 1 illustrates the morphological attributes of the *Cymbopogon flexuosus* (Steud.) Wats plant and focuses on the characteristics of trichomes and stomata.

Lemongrass cultivation is a rapidly growing economy. The major drive for lemongrass cultivation is its high industrial potential in the pharmaceutical, food, and cosmetics sectors. It is grown over an area of 16,000 ha on a global scale that produces about 1000 t of lemongrass essential oil (LEO) each year [8]. Out of this, India accounts for one-fourth of total production and area cropped worldwide [9]. Lemongrass cultivation can yield a net profit of about 300 USD per year per hectare [10]. A recent report by Plofshare Market Research pinned the global lemongrass oil market at 247 million dollars in 2019 and expects it to grow to 421 million dollars by 2027, at a compound annual growth rate of 7%. Additionally, lemongrass export of India has risen more than 1250% from 2001 to 2020, which solidifies its demand and importance (The Directorate General of Commercial Intelligence and Statistics, Kolkata, India). Moreover, the size of the essential oil market is estimated by numerous market research firms. One such study by Facts and Factors estimates its size to be 7 billion dollars, which is expected to double by 2027.

The present study aims to provide a timely and comprehensive review of the antimicrobial and anticancer potential of LEO, as well as to discuss the development of industrial applications based on both feasible and reproducible delivery systems.

## 2. LEO: Biosynthesis and Chemical Composition

Lemongrass (*Cymbopogon* spp.) oil is a cocktail of various terpenes and terpenoids, out of which the major components belong to cyclic and acyclic monoterpenes. The monoterpenes are derived from geranyl diphosphate (GPP). The GPP is a fusion component of isopentenyl diphosphate (IPP) and its allylic isomer dimethylallyl diphosphate (DMAPP) (Figure 2). The IPP is a precursor of all the terpenes and terpenoids. Earlier it was widely accepted that plants produce IPP through a cytoplasmic mevalonate (MVA) pathway. However, empirical studies reveal that a newly discovered methylerythritol phosphate (MEP) pathway for monoterpenes biosynthesis is more dominant in lemongrass species [11]. The plastidic MEP pathway begins with the reaction of pyruvate with thiamine diphosphate (TPP), which yields hydroxyethyl-TPP. Hydroxyethyl-TPP, upon reacting with glyceraldehyde 3-phosphate (GAP), releases 1-deoxy-D-xylulose 5-phosphate (DXP). Subsequently, DXP rearranges and reduces to MEP, which is further phosphorylated and ultimately generates IPP. 

On the other hand, the MVA pathway involves the condensation of three acetyl-CoA molecules into 3-hydroxymethylglutaryl-CoA (HMG-CoA). The HMG-CoA is reduced to MVA and the subsequent steps of MVA phosphorylation produce IPP. The IPP produced from the MEP/MVA pathway reacts with DMAPP to yield GPP, which subsequently forms geraniol in lemongrass. Geraniol, through different reversible reactions, produces all the major components of lemongrass oil [12] (Figure 2).

LEO is produced only in young and rapidly growing lemongrass leaves and floral tops and is stored in specialized parenchymal oil cells between vascular bundles proximal to non-photosynthetic tissue [13]. The amount of lemongrass oil is about 1–2% of the plant’s total dry weight [14]. However, various drying methods can produce different content and quality of LEO [15]. LEO composition may also vary with the extraction method, developmental stage of the plant, and the solvent used for extraction [16]. The major components in most lemongrass species include neral, isoneral, geranial, isogeranial, geraniol, geranyl acetate, citronellal, citronellol, germacrene-D, and elemol that make up about 60–80% of LEO [12,17] (Figure 3).

The remaining components also called minor components, comprise camphene, pinene, limonene, linalool, citronellyl acetate, elemene, and caryophyllene oxide [12]. The isomeric mixture of geranial and neral is known as citral and its content can be used as a quality marker for LEO [16]. This aldehyde monoterpene is the key active constituent (<80%) of *Cymbopogon flexuosus* oil, making it an aldehyde-type grass. Alternatively, *Cymbopogon martinii* (Palmarosa) has more alcohol content (nerol and geraniol) than aldehydes (neral and geranial) and, thus, is called an alcohol type grass, while *Cymbopogon winterianus* (Java citronella) is an intermediate type, due to the moderate contents of aldehyde and alcohol in its essential oil [18].

## 3. LEO Therapeutics

Recent decades mark an exponential upsurge in establishing the bioactivities of lemongrass extract and essential oil. Feeding studies establish that lemongrass oil and extract has antimicrobial, anticancer, antiamoebic, antidiarrheal, antifilarial, antitussive, antiseptic, larvicidal, insecticidal, miticidal, ovicidal, acaricidal, analgesic, aesthetic, anti-inflammatory, antioxidant, antinociceptive, antihypertensive, anti-obesity, anxiolytic, and antimutagenicity potential, cardioprotective, antirheumatic, and haematological properties [9,20,21,22,23,24,25,26,27,28,29,30,31,32,33]. These bioactivities are the direct product of the individual and synergistic effect of different major and minor LEO components [12]. LEO therapeutics is an emerging alternative for synthetic pharmaceuticals due to its naturality, biocompatibility, and inexpensiveness. The aim of this review is to summarize the state-of-art knowledge on the properties of LEO, the potential of applicative uses in fields such as agriculture, food preservation, veterinary medicine, human safety applications based on antibacterial and antifungal treatments of surfaces and prostheses, and on the anticancer effects that could exploit LEO components as adjuvants in association to standard chemotherapies.

### 3.1. Antimicrobial Potential

Lemongrass oil and extract are effective against a wide variety of disease-causing microbes. LEO has been used as an antibacterial [23,34,35], antifungal [32,36], and an antiviral [23,37,38] agent on many occasions. Similarly, lemongrass extract inhibited the growth of *Bacillus cereus*, *Escherichia coli*, *Klebsiella pneumoniae*, *Candida albicans*, and *Staphylococcus aureus* with different levels of susceptibility [39]. The EO components with different functional groups exhibit different levels of antimicrobial potential, where phenols and aldehydes have the highest activities, while esters and hydrocarbons have the least [40]. However, the antimicrobial activity of lemongrass is extensively attributed to the citral (aldehyde) present in its oil [28,41,42]. Yet, when a mixture of principal oil components and the whole EO were tested, the whole EO exhibited enhanced antimicrobial efficacy [43].

Singh et al. [43] studied 1114 strains of different microbes including moulds, yeasts, and bacteria from 29 genera and 105 species, and circled out about 425 LEO-sensitive microbial isolates. It is suggested that a low concentration of LEO inhibits microbial growth and development (bacteriostatic, fungistatic, and virustatic), while a higher concentration renders irreversible destruction leading to microbial death (bactericidal, fungicidal, and virucidal) [43,44]. Furthermore, one study addressed LEO antimicrobial potential against 42 microorganisms, including 20 bacteria, 15 fungi, and 7 yeasts [45].

In microbial studies, IC_50_ (half maximal inhibitory concentration) and MIC (minimum inhibitory concentration) values are two important markers that can be considered to determine the antimicrobial potential of any chemical. The IC_50_ of a drug is the concentration that can bring a 50% reduction in the microbial activity, therefore it may be cytostatic but bacteria can recover soon thereafter. In addition, concentrations are not always assessed precisely but are usually tested on a scalar dilution, from 1 to 10, and so on. However, it is more informative to know the MIC of a drug or plant extract, i.e., the concentration at which no visible growth of a microbe is detected. The MIC of LEO and citral against planktonic *Staphylococcus aureus* were noticed to be 0.0781% (*v*/*v*) and 0.0313% (*v*/*v*), respectively [44]. Similarly, LEO had higher efficacy (MIC—0.65 % *v*/*v*) against *Acinetobacter baumannii* strains over citral (MIC—0.14% *v*/*v*) [45]. This implies that its minor components, including limonene, linalool, and myrcene have specific as well as a synergistic mechanism with major components and can play a decisive role in augmenting oil effectivity [45,46,47]. In this regard, Table 1 addresses IC_50_ and MIC of *Cymbopogon citratus* (DC.) Stapf EO and citral against some common microbes as a variable for their antimicrobial efficacies. 

#### 3.1.1. Antibacterial Activity

The antibacterial characteristic of LEO is well established [18,40,49,50,51,52,53]. It has been suggested that LEO induces the destruction of bacterial biofilms and hinders further bacterial growth and development [54]. Furthermore, LEO components can destabilize the bonds between the lipid bilayer and neutralize the bacteria through membrane disintegration [55]. LEO can confer structural changes, as well, in different bacteria. It was reported to cause complete disfiguration and distortion in the *Pseudomonas* spp. [56]. The MIC values for LEO and citral against *P. aeruginosa* were calculated as >40% and 10%, respectively. Furthermore, LEO blocks biofilm formation in bacterial colonies [54], for example, 0.125% (*v*/*v*) of LEO can restrict biofilm formation in methicillin-resistant *Staphylococcus aureus* strains [57]. It can disrupt the cell membrane and inhibit cytoplasmic metabolism, making LEO effective against both Gram-negative and Gram-positive bacteria [34,49,55]. *Cymbopogon khasianus* essential oil inhibited the growth of *Escherichia coli* with the MIC and MBC (Minimum Bactericidal Concentration) values of 20 μg/mL each. It can also retard the growth of *Bacillus subtilis*, *Salmonella enterica typhimurium*, *Staphylococcus aureus*, *Klebsiella pneumoniae* with a MIC range of 25–50 μg/mL [47]. Multiple recent studies against MDR (multidrug-resistant) bacteria [58,59] show that, while a low concentration of LEO retards growth and biofilm formation, a higher concentration can confer complete elimination of *Salmonella* Heidelberg.

The bacteriostatic and bactericidal characteristics of LEO primarily depend on the bacteria and oil concentration [60,61]. However, several other factors, such as oil composition, extraction method, plant developmental stage, and environmental variables including temperature can influence the oil’s effectiveness. Therefore, lemongrass oils from different species might exhibit effects of different nature and intensity. Nevertheless, the host organism can also decide oil effectiveness to a certain extent depending on its morpho-physiological attributes [62]. Therefore, EOs react differently with Gram-positive and Gram-negative bacteria, owing to their dissimilar cell-wall structures [60,63]. Costa et al. [64] examined *Cymbopogon flexuosus* EO (μL mL_−1_) against *Listeria monocytogenes*, *Staphylococcus aureus,* and *Salmonella typhimurium* and determined their MICs and MBCs as of 3.9 μL mL_−1_ each. The MICs of citral against *Cronobacter sakazakii* strains ranged from 0.27 to 0.54 mg/mL. Scanning electron microscopy analysis further confirmed that *C*. *sakazakii* cell membranes were damaged by citral [65]. Table 2 traces the efficacies of lemongrass essential oils against various gram positive and gram negative bacteria.

#### 3.1.2. Antifungal Activity

The antifungal activity of LEO has been reported against multiple fungi [38,67,68,69]. Volatiles from lemongrass oil, such as phenols, flavonoids, and flavones, are effective against numerous fungal strains [70,71,72]. Helal et al. [73] reported that LEO caused plasma membrane disruption and disorganization of mitochondria and resulted in Ca^2+^, K^+,^ and Mg^2+^ leakage. The loss of ions can further affect signal transduction and fungal germination. Moreover, Alviano et al. [74] observed that LEO components induce cell size reduction and inhibit the spore germination in *Candida albicans.* LEO can directly act upon the fungal lipid bilayer owing to its readily volatile and lipophilic nature. It can form a charge–transfer complex with the lipid bilayer, destabilizing the membrane and inhibiting further membrane synthesis, and retards fungal spore formation and cellular respiration [74,75]. Boukhatem et al. [45] found that the vapor form of LEO inhibits mycotic growth and development more effectively than the liquid phase, probably because of the direct accumulation of LEO vapors on fungal mycelium. In this regard, Table 3 depicts the MIC values of EOs obtained from different *Cymbopogon* species against various fungal strains.

LEO components including citral, geraniol, myrcene, limonene, and linalool, have significant antifungal activity [30,38,40,47,67,73,74,75,76,77,78]. Geraniol increases the outward leakage rate of potassium ions, while citral damages the microtubules and exhibits cytotoxicity in fungi [27]. Linalool, a monoterpene alcohol, comprises numerous fungicidal properties [79]. It retards the overall development and propagation of different fungi through the respiratory restriction of their aerial mycelia [46]. Additionally, other aldehydes of LEO can confer antimycotic activity through cross-linkage reaction within the fungal membrane [46].

EOs can remain effective for a longer duration against fungal spore production, ensuring improved shelf life for food products [74,75,78]. It was also suggested that lemongrass oil induces reactive oxygen species (ROS) production in fungi and afflicts severe oxidative damage that leads to subsequent cellular death [32]. This can also enable EOs as a sustainable alternative in the food preservation and packaging industries [80,81,82]. Edible coating of EOs, including LEO on stored fruits, meat, and dairy products, discourages fungal attack and food spoilage through restricting fungal growth and reproduction [79,80,81,82,83]. The edible coatings of EOs have increased antimicrobial potential over the free EOs due to their altered surface charge and amplified action on multiple target sites in the mycotic membrane [84]. This extends edibility and maintains physicochemical qualities, including the tastes and odours of such products [76,85]. However, different EOs have different antimicrobial mechanisms, and thus acquisition of resistance by microbes against the wide array of compounds in EOs is rare [86].

#### 3.1.3. Antiviral Activity

In addition to its fungal and bacterial efficacy, LEO is also effective against numerous viruses. The antiviral activity of lemongrass oil was tested against herpes simplex virus-I (HSV-I) [36] and murine norovirus (MNV) [87]. The studies demonstrated that 0.1% and 2% of LEO concentration were sufficiently potent to inhibit the replication of HSV-I and MNV, respectively. Furthermore, LEO can weaken the HIV transcription and virus reactivation by interfering with the Tat/TAR-RNA complex. As the Tat protein enhances the efficiency of viral transcription, LEO’s interference with the Tat/TAR-RNA complex results in the downregulation of HIV activity [88]. Lemongrass volatiles were also able to cause more than 50% inhibition of the tobacco mosaic virus at a 100 µg/mL concentration [43]. Human mastadenovirus (HAdV) causes numerous ailments, such as respiratory infection, gastroenteritis, hepatitis, meningoencephalitis, pneumonia, and multiple others. Lemongrass extracts also induced cytotoxicity in a human lung adenocarcinoma cell line and monkey kidney cell lines and showed antiviral activity against HAdV [89]. In the recent pandemic upsurge, the efficacy of LEO was suggested against influenza and coronaviruses SARS-CoV-2, as well, enhancing the relevance and importance of lemongrass oil even more [37,89].

### 3.2. Antioxidant-Related Effects

Plant and animal cells produce various oxidative compounds, such as H_2_O_2_, O_2_^−^, and OH^−^, which can damage lipids, proteins, and DNA and can induce several health complications including cancer, aging, and neurological disorders in humans [26]. However, another group of compounds, known as antioxidants, has the potential to counter these effects [90,91]. Lemongrass possesses antioxidants that render protective measures against reactive species [92,93]. Lemongrass extracts have been reported to reduce reactive species concentration, lipid peroxidation, and decolourisation of 2,2-diphenyl-1-picrylhydrazyl [94,95]. Lemongrass extract can also buttress the endogenous antioxidant defence system in alveolar macrophages cells through augmenting superoxide dismutase activity and glutathione formation [95,96]. A vast array of plant extracts has been studied for their beneficial antioxidant properties. In particular, plant antioxidant potential may protect cells and organs from radical oxygen species. The natural extracts are usually compared to controls, such as butylated hydroxy anisol (BHA) or buthylhydrotoluene (BHT), for their antioxidant profile. Radical scavenging activity is evaluated by their activity towards a stable free radical, 2,2-azino-bis-3-ethylbenzothiazoline-6-sulphonic acid (ABTS). To date, there are no published reports on the antioxidant power of LEO. These data could be useful in comparing different cultivars or different methods of cultivation. Plant extracts have been studied, on cultured cells, for their antioxidant properties, their enzymes, and their redox potentials. In particular, regulation of redox status in cells may affect the levels of methyl group donor S-adenosylmethionine (SAM). SAM is a cofactor for histone methyltransferases and DNA methyltransferases. The consumption of glutathione, as occurs during oxidative stress, with an increase of its oxidized form, glutathione disulfide, may inhibit S-adenosylmethionine (SAM) synthetase, with a reduction of SAM synthesis [97] influencing the epigenetic modifications of proteins and DNA. Citral was shown to increase intracellular oxygen radicals, while inhibition of glutathione synthesis increased citral’s anticancer effect [97]. Citral was shown to modulate oxidative stress preferentially in cancer cells and to induce the endoplasmic reticulum stress exerting thus an antiproliferative action [98]. Lemongrass has a competitive advantage over other synthetic antioxidants, such as butylated hydroxytoluene, since they can induce haemorrhage: in this view, lemongrass oil is regarded as ‘safe’ for human consumption [98]. This opens a new vista for lemongrass oil in the food preservation and safety industries, including the meat, and dairy industries [46]. In the food industry, the oxidation of lipids is an important determinant of meat and dairy products. However, their highly rich nutritional profiles are prone to lipid peroxidation and quality deterioration. In this regard, coating such products with LEO minimizes lipid peroxidation and increases their shelf life and quality [99]. Furthermore, the antioxidant nature of citral is exploited in animal skin cancer models [100,101]. Soares et al. [102] reported that LEO, which was characterised in its major components, showed high antioxidant activity compared to the methanolic and aqueous extracts of lemongrass. In detail, they showed higher antioxidant capacity of LEO compared to the aqueous extracts of leaves: Using the 2,2-diphenyl-1-picrylhydrazyl free radical as a control, they obtained similar antioxidant power, at levels of 41 μg/mL. The antioxidant activity of LEO can further be enhanced through mixing it with other potent antioxidative agents. On this note, a mixture of LEO with *Ocimum gratissimum*, and *Thymus vulgaris* oil had enhanced effects against *Bipolaris oryzae* and *Alternaria alternata* [102,103]. A review on the antioxidant, antimicrobial and antifungal properties of LEO and recent updates on the possible applicative uses has been recently published [104].

### 3.3. Anticancer Activity

According to the World Health Organization (WHO), cancer caused an approximated 10 million deaths, or one in six deaths, in 2020. This situation is not going to be relieved, as there is estimated to be an increase of 45% in cancer mortality rate between 2008–2030. Among cancers, the most common types are breast cancer, lung cancer, colorectal cancer, prostate cancer, skin, and stomach cancer. The ongoing conventional chemotherapies, radiotherapy treatment, and surgeries have shown a large number of involuntary side effects due to insufficient knowledge of treatment specificity, and are not recommended for long-term usage [105,106].

Medicinal plants emerge as potential candidates in the cancer world and raise hopes for the scientific community. Scientists are constantly looking for natural sources to uncover the potential plant-based therapeutic agents having immense anticancer properties [20]. In this sense, the essential oil of lemongrass counts among such plants for its cytotoxicity on human cancer cells (Table 4). Its active ingredients, including geraniol, geranyl acetate, α-bisabolol, and iso-intermedeol have individually been found to impart cytotoxic effects on cancer cells [107]. Lemongrass EO has exhibited inhibition of human mouth epidermal carcinoma (KB) and murine leukemia cell lines (P388) [108].

Among LEO terpenes, citral, the major component of lemongrass oil, plays a potential role as antiproliferative against several types of cancer cells, such as the two human prostate cancer cell lines, LNCaP and PC-3 [107], HL60, U937 ovarian cancer cells [108,109,110], cervical cancer cell lines [111], and the breast cancer cell line, MCF-7 [112]. Interestingly, citral does not exert cytotoxicity to normal epithelial cells but exhibits toxic effects against human breast cancer cell lines, confirming its cancer-specific efficacy [113].

Although a large number of studies evinced the anticancer activity of lemongrass, scarce data are available on its mode of action. Studies based on different cancer cell types substantiated citral efficacy via activated procaspase-3, induction of apoptosis, and cell cycle arrest in the G2/M phase [114,115]. Citral consists of a double bond in conjugation with an aldehyde (α, β-unsaturated) group in its core structure, which serves as a potent caspase 3 activator, responsible for pro-apoptotic activity [114]. Moreover, citral-induced apoptotic activity was associated with DNA fragmentation and induced caspase 3 activity against hematopoietic cancer cell lines and ovarian cancer cell lines. Among the relationship between LEO constituents and intracellular signalling, several studies reported significant upregulation of antiapoptotic proteins.

The Src-tyrosine kinase is expressed in small cell lung cancer and can phosphorylate transcription factor Stat3(Y705) [114], which sequentially enhances the expression of downstream genes engaged in the antiapoptotic activity, i.e., Bcl-xL and Mcl-1 [116]. An experimental study showed the inhibitory effects of LEO and citral on phosphorylation of Src(Y416) blocking its activation, resulting in reduced phosphorylation of Stat3 (Y705). Non-phosphorylated Stat3 disrupts cell growth and the signal pathways that upregulate the expression of Bcl-xL and Mcl-1 [117]. Citral-dependent apoptosis induction has also been observed against prostate cancer cell lines. Citral induced gene activation initiates AMPK (an enzyme necessary in the fatty acid metabolism) phosphorylation resulting in the activation of BAX and the downregulation of Bcl-2, which initiates an apoptosis cascade in prostate cancer cell lines [118]. Citral-mediated breast tumour growth inhibition via the inhibition of ALDH1A3 was reported [119]. The up-regulation of retinoic acid (RA) signalling by ALDH1A3 can cause breast cancer growth, and citral inhibited the expression of RA-inducible genes mediated by ALDH1A3 [119]. Microtubule affinity regulating kinase 4 (MARK4), an AMP-activated protein kinase [120], is reported to mediate apoptosis, inflammation, and distinct regulatory pathways [121]. Alterations in MARK4 expression hamper the cell cycle and eventually cause cancer. Citral potentially binds to MARK4 and inhibits its kinase activity, and is being considered an effective strategy to prevent the growth of cancer cells and other MARK4 associated diseases [122]. Citral has been reported inducing the phosphorylation of p53 protein and the expression of Bax, while reducing the expression of the antiapoptotic factors Bc-2 and Bcl-xL in human colorectal cancer lines, i.e., HT116 and HT29 [123]. Citral interferes with the ERK1/2 pathway and reduces the translocation of ERK1/2 protein to the nucleus. There is a certain possibility of the involvement of ERK1/2 in melanoma carcinogenesis and the progression in presence of mutated *N-Ras* and *B-Raf*. Therefore, citral could negatively affect cancer growth by inhibiting the final step of the MAPK cascade [124].

Geraniol, the second major constituent of LEO, has garnered heed for its potentiality in cancer treatment. It has been reported that geraniol induces the production of ROS and inhibits the phosphorylation of tyrosine kinases, which, in turn, induce apoptosis of cancer cells [125]. Several studies have been conducted to gain insights into its anticancer activity [126,127,128]. Ornithine decarboxylase (ODC) plays a prime role in the synthesis of polyamines, providing stabilization to DNA structure [129]. A decrease in ODC activity after geraniol treatment has been observed in the intestinal adenocarcinoma Caco-2 cell line, which, in turn, caused DNA synthesis inhibition and cell cycle arrest in the S phase [129]. Polyamine metabolism is a potential target in the development of cancer-preventive drugs, therefore, geraniol mediated decline in ODC activity might have a useful clinical role [130]. Geraniol-induced inhibition of the proliferation of A453 and A549 human lung cancer cell lines has been reported. Geraniol alters the tubulin polymerization and disrupts the active property of both the studied cell lines, resulting in cell apoptosis. Geraniol arrested the G0/G1 phase in A431 cells, with no effects on sub-diploid cells, and the G2/M phase of A549 with the increased population of sub-diploid cells, in a dose-dependent manner. The inhibitory effects of geraniol might be interrelated with the observed alteration in the ODC activity [131]. Geraniol caused inhibition of cell cycle progression, exerting altered expression of cyclins D1, A, B1, CDK2, and the cyclin kinase inhibitor proteins p21 and p27 [132]. Geraniol has been reported to induce the expression of pro-apoptotic proteins Bcl-2, Bax, Bak, and caspase3/8/9 in several human cancer cell lines [128,133]. Moreover, the considerable increase in these proteins indicates that geraniol induces apoptosis through the mitochondrial intrinsic pathway [134]. The antiangiogenic activity of geraniol has been confirmed by both in-vivo and in-vitro studies. Geraniol suppresses the endothelioma cell line and reduces Ki67-positive cells and CD3-microvessela by suppressing the expression of VEGFR-2 in Balb/c mice [135]. This activity might play a role in reducing tumour growth, as the tumour needs a new blood vessel to grow. Geraniol arrested the proliferation of two pancreatic cancer cell lines, MIA PaCa-2 and BxPC, in hamsters when injected with PC-1 pancreatic ductal adenocarcinoma cells. In both the cell lines, it arrested the G1 phase of the cell cycle along with increasing the expression of the cyclin kinase inhibitor proteins p21^cip1^ and p27^kip1^ while suppressing those of cyclin A, B1, and CDK2 [136,137]. Figure 4 indicates a series of distinctive signalling pathways activated in cancer cells via different components of LEO.

D-limonene, another constituent of LEO, was also reported to possess antineoplastic activity. D-limonene enhances the activity of carcinogen metabolizing enzymes, such as cytochrome P450, responsible for the conversion of carcinogens into less harmful forms and blocks their interaction with DNA [138]. Treatment of D-limonene on LS174T human colon cancer cells inhibited the P13K/Akt pathway and induced cell apoptosis. An increase in PARP cleavage and activation of caspase-3 indicates the involvement of the mitochondrial apoptotic pathway [139]. Limonene-mediated induction of apoptosis via increased expression of Bax and caspase-3 and decreased Bcl-2 expression has been reported in T24 bladder cancer cells. Moreover, limonene arrested the cell cycle in the G2/M phase and wound healing and transwell assay using Matrigel has confirmed the limonene mediated suppression of cancer cell migration and invasion [140]. Another component of the EO of lemongrass, citronellol, is also found to exert cytotoxic effects on several cancer cell lines [141,142,143]. Citronellol showed its anticancer activity via increased reactive oxygen species production, alterations in mitochondrial permeability, DNA fragmentation, changes in cytochrome c activities, and activation of caspase, against the MCF-7 human mammary tumour cell line [144]. The cytotoxicity of α-bisabolol has been reported against human and rat malignant glioma cancer cell lines. In α-bisabolol-treated cell lines, the rapid loss of inner transmembrane potential and an increase in cytochrome-c translocation indicate that α-bisabolol can trigger apoptosis through mitochondrial intrinsic pathway [145]. Another experimental study has confirmed the cytotoxic effect of α-bisabolol against several cancer cell lines. α-bisabolol arrested cell cycle and initiated cancer cell death via a BID (BH3-only activator protein)-dependent mechanism [146]. It induces the permeability of the outer mitochondrial membrane and plays a crucial role during apoptosis [147]. α-bisabolol-induced damage to lysosomal and mitochondrial membranes via BID resulted in autophagy and regulated cell death, enlightening its modes of action [146,147,148,149].

All of the mentioned components of LEO have presented their chemo-preventative effects via arrest of different phases of the cell cycle, suppression of cyclins and cyclin-dependent kinases, DNA fragmentation, and antiangiogenic activity, against different cancer cell lines. Several distinct signalling pathways and mechanisms of action have been reported in experimental studies exhibiting the anticancer activities of these mentioned components (Table 5). 

Epigenetic modifications play a key role in cancer proliferation. Several chromatin remodelling complex components are found mutated, silenced, or overexpressed in cancers. Epigenetic mechanisms may be taken into account to explain the regulation of various protein-coding and protein non-coding genes. One type of regulation of gene transcription is based on the level of DNA methylation on promoters, controlled by DNA methyltransferases. A second mechanism is dependent on chromatin accessibility mediated by remodelling complexes, recruitment of Polycomb repressing complexes, and histone-modifying enzymes [97]. An involvement of non-coding RNAs (nc-RNAs) is at the basis of these mechanisms; therefore, the expression of long and small non-coding RNAs determines whether an antioncogenic pathway is repressed or downregulated, or oncogenes are set free to induce cell transformation. Therefore, chromatin accessibility (opening or compaction), access to transcription machinery, and promoter methylation are the principal mechanisms that are targeted by plant extracts, essential oils, and individual bioactives. Various classes of bioactives were shown to exert anticancer effects through the downregulation of microRNAs, most often oncomiRs, and the release of mRNAs coding for antiproliferative proteins. The review by Sabo [152] and that by Cherng [153] showed the relationship between natural compounds and nc-RNAs in cancer cells, and the potential use of their bioactives in cancer therapies. Xu and colleagues described, for the first time, the involvement of pinene, one component of LEO, in the upregulation of p27/CDKN1B, a cell cycle-blocking protein, through the downregulation of miR-221 [153]. The most well-known group of nc-RNAs are microRNAs [154]. When they are abundant, they silence mRNA transcription by sequestering them and destining them to degradation. Long non-coding RNAs, such as competing endogenous RNAs (ceRNAs), may sponge a group of miRNAs and the relative abundance thereof determines whether the miRNA can exert its effects or is bound to the sponge [154]. This has been clearly reviewed for stilbenes [155] and other plant bioactives [155,156].

It should be taken into consideration that often individual components were shown to be not effective as the essential oil, possibly because the association of multiple components potentiates the activity of each molecule. LEO effects on doxorubicin-resistant ovarian carcinoma cells were shown not dependent on citral [46]. The natural mixture of bioactives present in LEO is responsible for the beneficial effects, such as the regulation of multidrug resistance and P-glycoprotein efflux pump inhibition in human ovarian carcinoma cells [46] and colon cancer cells [157].

Furthermore, other studies showed the anticancer effect of LEO, while anticancer activity could not be attributed to its single constituents [158,159]. The anticancer activities of plant essential oils have been usually used as a paragon to compare essential oils from different sources. For instance, zingiber essential oil (ZEO) components nerol, citral, limonene, pinene, and camphene sum to 55–60% of ZEO bioactives. ZEO was shown to be active against colorectal cancer in rats [160]. On the other hand, when specific bioactives have been studied as anticancer agents, pleiotropic effects have been observed, without precise indication of the main target or signalling pathway. For instance, resveratrol and stilbenes are known to regulate different signalling pathways [154]. The biochemical interaction with enzymes, regulation of non-coding RNAs, and activation of signalling pathways and transcription factors are among the main effects observed. 

In addition to essential oils, other lemongrass extracts have been tested for anticancer activity. For instance, LEO ethanolic extracts showed an anticancer effect on cancer cells, and an increased level of ROS was shown to affect the increase in apoptosis levels [159]. Although these extracts have a complex composition, some of the components may directly derive from the essential oil. Therefore, an analytical procedure is required to define these compositions and to discern whether their anticancer effects derive from one compound or the cumulative effect of several compounds.

### 3.4. Miscellaneous

Lemongrass tea relieves stress and removes cough and nasal congestion. LEO is exploited in the production of various mouthwashes [161]. The distinct fragrance of lemongrass is exploited in the flavour and perfume industries. Asian cuisine has long used lemongrass in numerous traditional dishes’ preparations. LEO coating on berries slows down the increase in total anthocyanin concentration, a known marker for the ripening of berry fruits, thus delaying berry ripening and spoilage [162]. Lemongrass oil and its components, such as citral, myrcene, and citronellol, show antimalarial potential against *Plasmodium* spp. as well as confer total mortality of stage III and stage IV larvae in *Anopheles funestus s.s* [163,164,165]. Furthermore, multiple reports suggest LEO can suppress larval growth and production in various insects [62,166,167,168,169,170,171]. The lemongrass active constituent, citral, in combination with other components can regulate neuroreceptor activities, signal transduction, hormonal balance, membrane integrity, and cytotoxicity in insects [172,173]. Other minor components, including caryophyllene, caryophyllene oxide, and germacrene-D, discourage insect invasion and are effective against common houseflies and mosquitoes as well [174]. LEO can retard the activities of different neurotransmitters such as acetylcholine esterase and octopamine, activate olfactory receptor neurons, and induce related neurotoxic responses in insects [11,175,176]. Concerning this, a recent review [11] can be referred to for further reading on the insecticidal property of LEO and its underlying mechanism.

## 4. Perspectives

Although recent years have seen an upsurge in lemongrass production, it is still far behind the global demand. Given its potential applicability in diverse sectors, lemongrass farming requires a revolutionized future. Harvesting of plant metabolites from controlled growth conditions (hairy root cultures, greenhouse) may establish a standard production of individual LEO bioactive components. Moreover, since citral is considered as the key marker for lemongrass quality and medicinal potential, improved lemongrass varieties with higher citral content are the need of the hour. In these paragraphs, the bioactive LEO phytocomponents were introduced for a myriad of medicinal properties including antimicrobial, anticancer, antioxidant, insecticidal, and antimalarial activities. Although LEO and its constituents have shown anticancer activity in vitro, and, in some cases, in animal studies, only a few researchers to date have tested the delivery of its bioactive components combined with nanoparticles or delivery systems [150,177]. This is required in order to bypass the metabolism of bioactives by microbiota and by cellular enzymes, to provide stability and bioavailability at the active concentrations tested in these studies. The anticancer effect of lemongrass extracts on cancer cells having an increased level of ROS was shown to affect an increase in apoptosis levels [161]. Thus, it is possible that different bioactive contents may be established for their relationships with different activities against cancer cells. In addition, a variety of applications, from food safety and food preservation, in terms of antioxidant potential as well as for antifungal properties, to applications in agriculture and veterinary medicine, and as coatings on biopolymers for surgery (maxillofacial silicone specimens in dentistry, other medical implants) have been recently proposed. The ability of LEO terpenes to stop bacteria and fungi from growing in biofilms [178,179,180,181,182,183] has also a wide array of applicative uses in medicine and surgical devices, and in industrial solutions to biocorrosion, biofouling, biodegradation, water microbiology, and the control of bacterial quorum sensing signals [103]. The ability to modify the materials used in medical devices allows the application of LEO components to make such surfaces resistant to biofilm formation. The technological development of LEO applications is directed to the potentiation of its bioactivity through extended activity by combination with other bioactives, or by stabilization through nanoemulsions, nanoparticles, and polymers to coat surfaces. There are still obstacles to be overcome in reaching industrial applicative uses since polymer micro- and nano-capsules manufacture requires the control of compartmentalization on the micro and/or nano scales [184]. As for the present, technology, transfer, and scale-up applications are directed to high-priced products, while for low-income countries, LEO still may be an easy-to-use product with low-cost applicative use in traditional medicine and agriculture.

## 5. Conclusions

LEO has been recognized as a natural mixture, rich in powerful bioactives, that may have applications as antibacterial, antifungal, anthelmintic, for the storage and preservation of food quality, in agriculture, and animal health. LEO’s beneficial effects on normal cells combined with anticancer activity may open new pathways for the treatment of cancers. Further studies on the antioxidant and anticancer mechanisms exerted by lemongrass components are required, in vitro and in vivo, to validate these preliminary data.

## Figures and Tables

**Figure 1 antioxidants-11-00020-f001:**
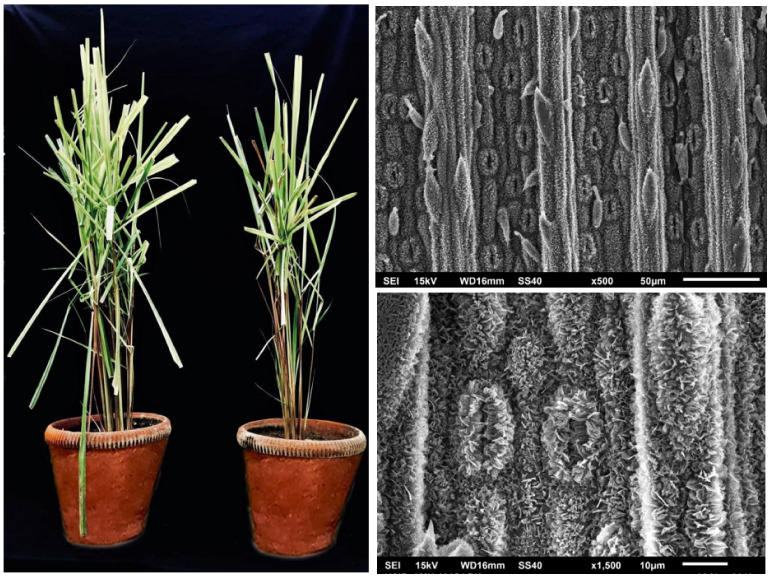
Morphological and anatomical profile of lemongrass (*Cymbopogon flexuosus* (Steud.) Wats) plants. The morphological representation of 150-days-old healthy lemongrass plants. Scanning electron microscopy (SEM) analysis of lemongrass leaf visualizing trichomes at ×500 magnification, stomatal characteristics at ×1500 magnification and Images provided by Mukarram’s laboratory.

**Figure 2 antioxidants-11-00020-f002:**
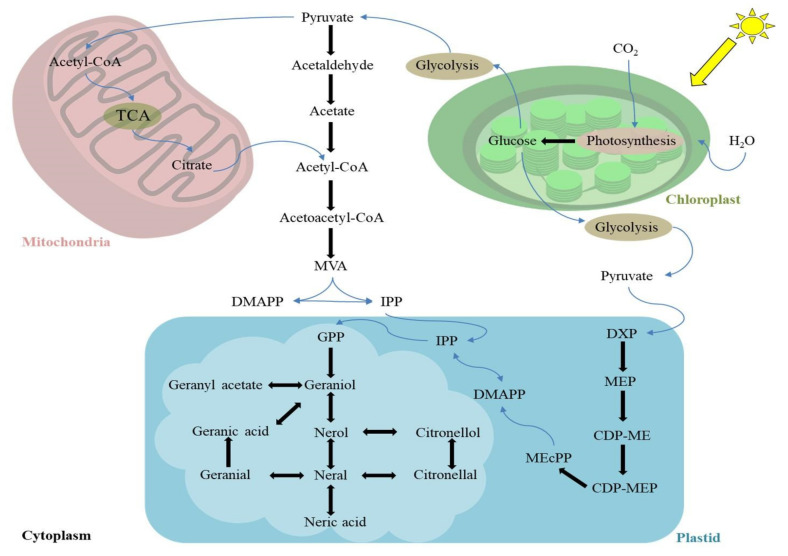
A mechanistic model for the biosynthesis of lemongrass essential oil and its crosstalk with other metabolic processes. Different cellular organelles work in tandem for oil production in lemongrass leaves. The lemongrass chloroplast, like most of the other plants, produces glucose through photosynthesis. The glucose undergoes glycolysis in the cytoplasm and yields pyruvate, a two-carbon compound. Lemongrass uses pyruvate as a substrate for the biosynthesis of isopentenyl diphosphate (IPP) units, either through the cytoplasmic mevalonate (MVA) pathway or plastidic methylerythritol phosphate (MEP) pathway in their young and rapidly growing leaves. Alternatively, mitochondria can import pyruvate and yield citrate through the tricarboxylic acid (TCA) cycle. The citrate can transform into Acetyl-CoA and join the MVA pathway to yield IPP units. The IPP produced through both pathways is converted into geraniol mediated by geranyl diphosphate (GPP) in lemongrass plastids. The geraniol is considered as a precursor for essential oil biosynthesis in lemongrass and yields all the major components through multiple reversible and irreversible reactions. The plastidic bubble highlights these reactions along with their substrates. DMAPP, dimethylallyl diphosphate; DXP, 1-deoxy-D-xylulose-5-phosphate; MEP, 2-C methylerythritol 4-phosphate; CDP-ME, 4-diphosphocytidyl-2-C-methyl-D-erythritol; CDP-MEP, 4-diphosphocytidyl-2-C-methyl-D-erythritol-2-phosphate; MEcPP, 2-C-methyl-D-erythritol 2,4-cyclodiphosphate. The image is drawn by the authors on the basis of teaching texts for plant biochemical pathways for secondary metabolites.

**Figure 3 antioxidants-11-00020-f003:**
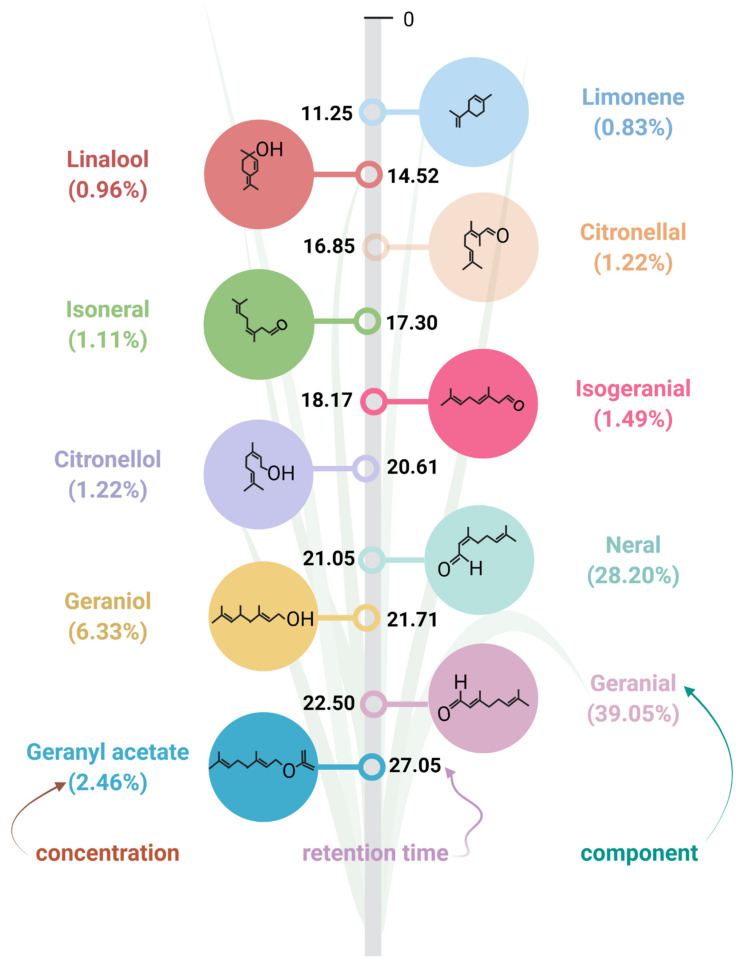
The abundant bioactive phytocomponents present in lemongrass essential oil. *Cymbopogon flexuosus* (Steud.) Wats var. Nima was grown and cultivated in Aligarh, India (27°52′ N, 78°51′ E). The fresh plant leaves were cultivated, and oil was extracted according to Guenther [19]. Further, gas chromatography-mass spectrometry (GC-MS) analysis revealed the compositional makeup of LEO. LEO components were distinguished according to their retention times (depicted, here, only from 0 to 27.05) and are mentioned here following their order of elution in the solid phase. Percentage abundance below each compound represents the percent peak area covered by the component during GC-MS analysis. Created with Biorender.com.

**Figure 4 antioxidants-11-00020-f004:**
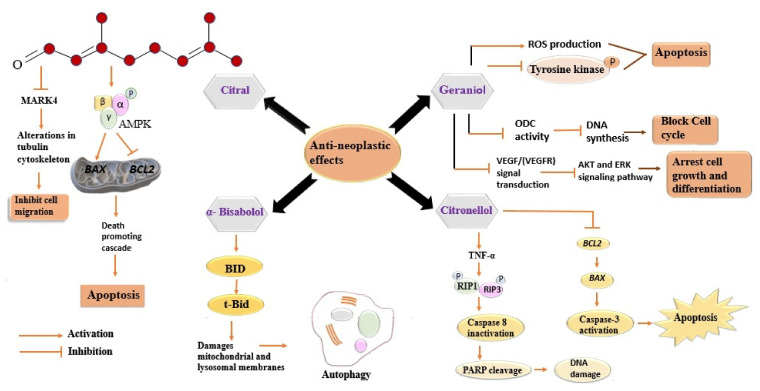
Distinctive signalling pathways activated in cancer cells via different components of LEO. Every component acts differently against cancer cells and involves diverse signalling pathways. All the involved pathways lead to inhibition of cell migration, cell cycle, and DNA synthesis. All these events eventually cause cell death (apoptosis). MARK4, Microtubule affinity-regulating kinase 4; AMPK, 5′ adenosine monophosphate-activated protein kinase; BAX, BCl2- associated X protein; BCL2, B-cell lymphoma 2; BID, BH3-only activator protein; tBid, truncated Bid, ROS, reactive oxygen species; ODC, ornithine decarboxylase; VEGF, vascular endothelial growth factor; VEGFR, vascular endothelial growth factor receptor; AKT, Ak strain transforming; ERK, extracellular regulated kinase; TNF-α, tumour necrosis factor; RIP1, receptor-interacting serine-threonine protein kinase 1; RIP3, receptor-interacting serine-threonine protein kinase 3; PARP, poly ADP ribose polymerase; DNA, deoxyribonucleic acid.

**Table 1 antioxidants-11-00020-t001:** Antimicrobial activity of *Cymbopogon citratus* (DC.) Stapf essential oil and citral. MIC, minimum inhibitory concentration; IC_50_, half maximal inhibitory concentration. Adapted from Viktorová et al. [48].

Species	Microbe	Family/ Kingdom	*Cymbopogon citratus* Essential Oil		Citral	
			IC_50_ [µL/L]	MIC [µL/L]	IC_50_ [µL/L]	MIC [µL/L]
*Candida famata*	fungi	Saccharomycetaceae	177 ± 19	3684 ± 271	37 ± 7	142 ± 19
*Cryptococcus albidus*	fungi	Tremellaceae	199 ± 25	265 ± 31	2 ± 0	20 ± 6
*Candida albicans*	fungi	Saccharomycetaceae	571 ± 109	2734 ± 250	83 ± 8	110 ± 15
*Mycobacterium smegmatis*	bacteria	Mycobateriaceae (Gram positive)	860 ± 89	3409 ± 775	109 ± 12	137 ± 19
*Proteus vulgaris*	bacteria	Enterobacteriaceae (Gram negative)	992 ± 37	1453 ± 40	97 ± 12	163 ± 34
*Staphylococcus aureus*	bacteria	Staphylococcaceaea (Gram positive)	1841 ±199	5830 ± 198	77 ± 2	92 ± 2
*Pseudomonas aeruginosa*	bacteria	Pseudomonadaceae (Gram negative)	2385 ± 162	5308 ± 339	41 ± 2	93 ± 8
*Salmonella enterica*	bacteria	Enterobacteriaceae (Gram negative)	2626 ± 301	4693 ± 634	66 ± 8	97 ± 3

**Table 2 antioxidants-11-00020-t002:** The antibacterial potential of essential oil obtained from different lemongrass species against common pathogenic bacteria. MIC, minimum inhibitory concentration; MBC, minimum bactericidal concentration.

Test Organisms	Family	Lemongrass Essential Oil			References
		Species	MIC	MBC	
*Staphylococcus aureus*	Staphylococcaceae	*C. citratus*	0.06 *	0.06 *	[34]
*Bacillus cereus*	Bacillaceae	*C. citratus*	0.06 *	0.06 *	
*Bacillus subtilis*	Bacillaceae	*C. citratus*	0.06 *	0.12 *	
*Escherichia coli*	Enterobacteriaceae	*C. citratus*	0.12 *	0.12 *	
*Klebsiella pneumoniae*	Enterobacteriaceae	*C. citratus*	0.50*	0.50 *	
*Staphylococcus aureus*	Staphylococcaceae	*C. flexuosus*	0.0781 *	-	[24]
*Enterococcus faecalis*	Enterococcaceae	*C. giganteous* and *C. citratus*	1 **	-	[66]
*Staphylococcus aureus*	Staphylococcaceae	*C. giganteous* and *C. citratus*	2.5 **	-	
*Listeria monocytogenes*	Listeriaceae	*C. giganteous* and *C. citratus*	8.3 **	-	
*Enterobacter aerogenes*	Enterobacteriaceae	*C. giganteous* and *C. citratus*	13.3 **	-	
*Escherichia coli*	Enterobacteriaceae	*C. giganteous* and *C. citratus*	10 **	-	
*Pseudomonas aeruginosa*	Pseudomonadaceae	*C. giganteous*	>80 **	-	
*Salmonella enterica*	Enterobacteriaceae	*C. giganteous* and *C. citratus*	2.1 **	-	
*Salmonella typhimurium*	Enterobacteriaceae	*C. giganteous* and *C. citratus*	2.5 **	-	
*Shigella dysenteriae*	Enterobacteriaceae	*C. giganteous* and *C. citratus*	8.3 **	-	
*Escherichia coli*	Enterobacteriaceae	*C. khasianus*	20 ^#^	20	[43]
*Staphylococcus aureus*	Staphylococcaceae	*C. khasianus*	30 ^#^	–	
*Pseudomonas aeruginosa*	Pseudomonadaceae	*C. khasianus*	20 ^#^	30	
*S. enterica typhimurium*	Staphylococcaceae	*C. khasianus*	30 ^#^	–	
*Bacillus subtilis*	Bacillaceae	*C. khasianus*	25 ^#^	–	
*Klebsiella* *pneumoniae*	Enterobacteriaceae	*C. khasianus*	20 ^#^	–	

* (*v*/*v*%), ** (mg/mL), ^#^ (μg/mL).

**Table 3 antioxidants-11-00020-t003:** The efficacy of essential oils obtained from different lemongrass species against some common pathogenic fungi. MIC, minimum inhibitory concentration.

Fungal Species	Family/Kingdom	Lemongrass Essential Oil		References
		Species	MIC	
*Candida albicans*	Saccharomycetaceae	*C. flexuosus*	0.0781 *	[26,77]
*Candida tropicalis*	Saccharomycetaceae	*C. flexuosus*	0.039 *	
*Candida albicans*	Saccharomycetaceae	*C. khasianus*	100	[43]
*Aspergillus flavus*	Trichocomaceae	*C. citratus*	50 ^##^	[45]
*Aspergillus niger*	Trichocomaceae	*C. citratus*	30 ^##^	
*Alternaria alternata*	Pleosporaceae	*C. citratus*	30 ^##^	
*Aspergillus fumigatus*	Trichocomaceae	*C. citratus*	30 ^##^	
*Fusarium moniliforme*	Nectriaceae	*C. citratus*	30 ^##^	
*Cochliobolus lunatus*	Pleosporaceae	*C. citratus*	30 ^##^	
*Drosera indica*	Droseraceae	*C. citratus*	20 ^##^	

* (*v*/*v*%), ## (ppm).

**Table 4 antioxidants-11-00020-t004:** Efficacy of essential oils obtained from different lemongrass species against various cancer cell lines. IC_50_, half maximal inhibitory concentration.

Tissue	Cell Lines	Lemongrass Essential Oil		References
		Species	IC_50_ (μg/mL)	
colon	HT-29	*C. flexuosus*	42.4	[107]
	HCT-152	*C. flexuosus*	60.2	
	SW-620	*C. flexuosus*	28.1	
	502713	*C. flexuosus*	4.2	
lung	H-226	*C. flexuosus*	61.4	
	A-549	*C. flexuosus*	49.7	
	Hop-62	*C. flexuosus*	79	
liver	Hep-2	*C. flexuosus*	4.8	
cervix	SiHa	*C. flexuosus*	6.5	
prostate	DU-145	*C. flexuosus*	41.5	
oral	KB	*C. flexuosus*	50.8	
neuroblastoma	IMR-32	*C. flexuosus*	4.7	
lung	A549	*C. citratus*	1.73	[107]
	H1975	*C. citratus*	4.01	

**Table 5 antioxidants-11-00020-t005:** Effects of different components of lemongrass essential oil on various cancer cell lines.

Components	Experimental Model	Mechanism of Action	References
citral	A549 (human lung carcinoma)	Growth arrest of cell cycle at sub G1 phase	[115]
	NCI-H1975 (human lung adenocarcinoma)	Up-regulation of procaspase-3	
	NCI-H1650 (human lung adenocarcinoma)	Decrease of Bcl-2 and increase of expression of Bax	
	NCI-H1299 (human lung large cell carcinoma)		
citral	Prostate cancer cells PC3 and PC3M (metastatic)	Inhibition of colony formation, suppression of expression of AMPK pathway genes *SREBP1*, *ACC*, *HMGR*	[119,150]
	Colony forming assay 10, 15, 25, 50, 100 μg/Ml	216 upregulated genes 396 downregulated genes	
	ATCC-CRL-1739/	Apoptosis, block of colony formation and migration	
	AGS stomach cancer cells		
	5, 10, 20 μg/mL		
citral	Human colorectal cancer HCT116 and HT29 cells	Induction of phosphorylation of p53, triggering ROS mediated mitochondrial intrinsic apoptosis	[124,151]
	HT29, SW620 lines	Cytotoxicity	
geraniol	A549 human lung adenocarcinoma cells in culture and in vivo in nude mice	Decreased the level of membrane-bound Ras protein, decreased the level of cholesterol and HMGCR protein	[127]
geraniol	In vitro murine endothelial-like eEND2 cells and HDMEC (dermal microvascular endothelial cells),	Blocked VEGF/ VEGFR signal transduction and suppression of cAKT and ERK signalling pathways	[128]
	In vivo, CT26 cell lines from undifferentiated colon carcinoma of the BALB/ c mouse		
geraniol	Human hepatoma (HepG2) and human lung adenocarcinoma (A549) cell lines	Growth arrest in G0/G1 interphase of the cell cycle, increased the production of ROS	[129]
citronellol	Invitro, non-small lung cancer cell (NCI-H1299);	Arrest of cell cycle at G1 phase, down-regulation of expression of cyclin E, and cyclin D, increase in expression of TNF-α, and activation of RIP1/RIP	[129]
	In vivo, injected NCI-H1299 into BALB/c nude mice		
citronellol	Triple-negative breast cancer MDA-MB-231 cell line	Decreased the expression of Bcl-2 gene and protein and increased Bax expression.	[141]
citronellol	DMBA(7,12-dimethylbenz(*a*) anthracene) induced mammary cancer in rats	Down-regulation of expression of NF-kB, IL-6, and TNF-α. Suppression of activity of COX-2.	[142]
α-bisabolol	CML-T1, Jurkat, HeLa cell lines	Cytotoxicity via mitochondria and lysosome-initiated caspase cascade and induction of autophagy and apoptosis	[143]
α-bisabolol	KLM1, KP4, and Panc 1 human pancreatic cancer cell lines	Up-regulation of *KISS1R*	[146]
α-bisabolol	Endometrial cancer cell lines RL95–2, ECC001, ECC003	Decreased activity of COX-2, induction in PARP cleavage, increased apoptosis via XIAP/ caspase 3 pathway	[148,149]
	Ishikawa cell line		
	ECC E6/E7 cell line		
limonene	Bladder cancer line T24;	Arrest of cell cycle in the G2/M phase; block of cancer cell migration; apoptosis; inhibition of PI3K/AKT pathway induces cell cycle G2/M, suppressing migration. Induces chromatin concentration, nuclear fragmentation, increases Bax, caspase 3, decreases Bcl-2	[135,149]
	colon cancer LS174T line;		
	Bladder cancer cells		
pinene	HepG2, HCC cells	CDKN1C/p57 and p27/CDKN1B upregulation, miR-221 downregulation	[152,153,154]
linalool	HeLa,	antiproliferative	[150]
	H520 lung cancer line,		
	BCC-1/KMC skin cancer		

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
