# Peer review of "Lemongrass Essential Oil Components with Antimicrobial and Anticancer Activities"

_antioxidants, 2021, doi:10.3390/antiox11010020_

Round 1
Reviewer 1 Report
The reviewed paper summarizes the knowledge about the chemical composition and biological activity of lemongrass essential oil. There is quite large amount of literature on the topic, but the comprehensive review of present knowledge is necesasary. The paper is quite interesting. However, in my opinion, it needs some minor corrections before publication. The list of my notes are below.
- There is lack of clear written aim of the paper. The Authors should strongly underline the importance of the described topic
- There is lack of citation in the legend of Figures 1, 2 and 3.
- Lines 133-137 it should be unified the way of writing the activity of the essential oils (for example cardio-protective or cardioprotective, etc.)
- Line 210 Is the word: „agaiJunst” written correctly?
- List of references should be revised by the Authors very carrefully and written in the same style. There are some lines where the style of citation is different (for example lines 571-2; 593-5). The latin names of the species should be written in italycs (for example lines 647-8) in the whole references list.
To conclude, in my opinion, the paper may be published in „Antioxidants” after minor revision.
Author Response
Thank you for kind and ensigthful remarks. We believe we addressed all the requests.
"There is lack of clear written aim of the paper. The Authors should
strongly underline the importance of the described topic"
Authors:
Yes. We added a sentence in introduction
The present study aims to provide a timely and comprehensive review of the antimicrobial and anticancer potential of LEO, as well as to discuss the development of industrial applications based on delivery systems feasible and reproducible.
And also, a the end of paragraph 3.0
Aim of this review is to summarize the state-of-art knowledge on the properties of LEO, the potential of applicative uses in fields such as agriculture, food preservation, veterinary medicine, human safety applications based on antibacterial and antifungal treatments of surfaces and prostheses, and on the anticancer effects that could allow to exploit LEO components as adjuvants in association to standard chemotherapies.
and also a perspective paragraph before conclusion
Ciations to figure were added, and all correcions were made.

Reviewer 2 Report
The review is very interesting and very good structurate. I suggest few minor language and editing corrections.
Author Response
Thank you for statement. We further implemented the review.
Reviewer 3 Report
The review “Lemongrass Essential Oil Components with Antimicrobial and Anticancer Activities antioxidants” deals with a remarkable subject on the area of the EO. The topic is very interesting for the definition of the Lemongrass applications in several fields as therapeutic compounds.
Overall, this is a well-organized paper, the review is significant and contributes useful information, but the manuscript would benefit from some little suggestions.
Taking in consideration the topics of Antioxidant journal, the review fits with the specific scope: natural and synthetic antioxidants and their relevance to plant, animal and human health and disease.
However, the part of Antioxidant related effects is very short (lines 267-308). I suggest to move this part after the 2.LEO: Biosynthesis and chemical composition.
Authors can stress additional information about the antioxidant activity and its interaction/influence with antimicrobial and anticancer activities.
Please consider the comments below
Minor comments
Line 113: please avoid: The predominant LEO components are terpenes or their derivatives. This sentence is redundant.
Table 1; I’m confused, this is a scheme (figure) not a table. Please check it!
Line 163; the sentence is not clear. Please re-formulate the definition of IC50.
In general, the citrated articles are recent and the resuming tables are clear. I kindly suggest, if it is possible, to include comments and some additional information about the technology development of the LEO application and, if there are some examples of technology transfer and scale-up application.
This information can increase the level of the review; however, this is only a suggestion.
Author Response
We thank the reviewer for criticism and we appreciate the suggestions made.
"the part of Antioxidant related effects is very short (lines
267-308). I suggest to move this part after the 2.LEO: Biosynthesis and
chemical composition.
Authors can stress additional information about the antioxidant activity and its interaction/influence with antimicrobial and anticancer activities."
We have added several new senences to adddress the remarks
Although LEO and its constituents have shown anticancer activity in vitro, and in some cases in animal studies, only a few researchers up to now tested the delivery of its bioactive components combined with nanoparticles or delivery systems [119,150,178]. This is required in order to bypass the metabolism of bioactives by microbiota and by cellular enzymes, to provide stability and bioavailability at the active concentrations tested in the studies. The anticancer effect of lemongrass extracts on cancer cells having an increased level of ROS was shown to affect the increase in apoptosis levels [162]. Thus, it is possible that different bioactive contents may be put in relationship with different activities against cancer cells. In addition, a varieties of applications, from food safety and food preservation, as for antioxidant potential as well as for the antifungal properties, to applications in agriculture and in veterinary medicine, and as coatings on biopolymers for surgery (maxillofacial silicone specimens in dentistry, other medical implants) have been recently proposed. The ability of LEO terpenes to stop bacteria and fungi to grow in biofilms has also a wide array of applicative uses in medicine and in surgical devices, and in industrial solutions to biocorrosion, biofouling, biodegradation, water microbiology, and control of bacterial quorum sensing signals [103]. The ability to modify the materials used in medical devices allow application of LEO components to make the surfaces resistant to biofilm formation. technology development of the LEO application is directed to potentiation of the bioactivity through extended activity by combination with other bioactives, or by stabilization through nanoemulsions, nanoparticles, and polymers to coat surfaces. There are still obstacles to be overcome to reach industrial applicative uses, since polymer micro- and nano-capsules manufacture requires control of compartmentalization on the micro and/or nano-scale. As for the present, technology transfer and scale-up application are directed to the high-priced products, while for low income countries LEO still may be an easy to use product with low cost applicative use in traditional medicine and in agriculture. the
6. CONCLUSIONS
LEO has been recognized as a natural mixture rich in powerful bioactives, that may have applications as antibacterial, antifungal, antihelmtinic, for storage and preservation of food quality, in agriculture, and in animal health. The beneficial effects on normal cells combined with anticancer activity may open new pathways for treatment of cancers. Further studies on the antioxidant and anticancer mechanisms exerted by lemongrass components are required either in vitro as well as in vivo, to validate these preliminary data.
Please consider the comments below
Minor comments
Line 113: please avoid: The predominant LEO components are terpenes or
their derivatives. This sentence is redundant.
Table 1; I'm confused, this is a scheme (figure) not a table. Authors:
checked, renamed!
Line 163; the sentence is not clear. Please re-formulate the definition
of IC50.
thank you. We provide a revised vwrsion with yellow labels on new sentences and corrections
In general, the citated articles are recent and the resuming tables are clear. I kindly suggest, if it is possible, to include comments and some additional information about the technology development of the LEO application and, if there are some examples of technology transfer and scale-up application.
Although LEO and its constituents have shown anticancer activity in vitro, and in some cases in animal studies, only a few researchers up to now tested the delivery of its bioactive components combined with nanoparticles or delivery systems [119,150,178]. This is required in order to bypass the metabolism of bioactives by microbiota and by cellular enzymes, to provide stability and bioavailability at the active concentrations tested in the studies. The anticancer effect of lemongrass extracts on cancer cells having an increased level of ROS was shown to affect the increase in apoptosis levels [162]. Thus, it is possible that different bioactive contents may be put in relationship with different activities against cancer cells. In addition, a varieties of applications, from food safety and food preservation, as for antioxidant potential as well as for the antifungal properties, to applications in agriculture and in veterinary medicine, and as coatings on biopolymers for surgery (maxillofacial silicone specimens in dentistry, other medical implants) have been recently proposed. The ability of LEO terpenes to stop bacteria and fungi to grow in biofilms has also a wide array of applicative uses in medicine and in surgical devices, and in industrial solutions to biocorrosion, biofouling, biodegradation, water microbiology, and control of bacterial quorum sensing signals [103]. The ability to modify the materials used in medical devices allow application of LEO components to make the surfaces resistant to biofilm formation. technology development of the LEO application is directed to potentiation of the bioactivity through extended activity by combination with other bioactives, or by stabilization through nanoemulsions, nanoparticles, and polymers to coat surfaces. There are still obstacles to be overcome to reach industrial applicative uses, since polymer micro- and nano-capsules manufacture requires control of compartmentalization on the micro and/or nano-scale. As for the present, technology transfer and scale-up application are directed to the high-priced products, while for low income countries LEO still may be an easy to use product with low cost applicative use in traditional medicine and in agriculture. the
6. CONCLUSIONS
LEO has been recognized as a natural mixture rich in powerful bioactives, that may have applications as antibacterial, antifungal, antihelmtinic, for storage and preservation of food quality, in agriculture, and in animal health. The beneficial effects on normal cells combined with anticancer activity may open new pathways for treatment of cancers. Further studies on the antioxidant and anticancer mechanisms exerted by lemongrass components are required either in vitro as well as in vivo, to validate these preliminary data.
This manuscript is a resubmission of an earlier submission. The following is a list of the peer review reports and author responses from that submission.